# Infestation of Early- and Late-Flushing Trees by Spring Caterpillars: An Associational Effect of Neighbouring Trees

Lenka Sarvašová [1,*] , Peter Zach [1], Michal Parák [2], Miroslav Saniga [1] and Ján Kulfan [1]

[1]  Institute of Forest Ecology, Slovak Academy of Sciences, Ľ. Štúra 2, 960 53 Zvolen, Slovakia; zach@ife.sk (P.Z.); miro.saniga@gmail.com (M.S.); kulfan@ife.sk (J.K.)
[2]  Independent Researcher, Banšelova 28, 821 04 Bratislava, Slovakia; parcius23@gmail.com
[*]  Correspondence: lenka.sarvasova14@gmail.com

**Abstract:** In temperate forests within Europe, early-flushing (EF) deciduous trees are often heavily infested by early spring leaf-eating Lepidoptera, while late-flushing (LF) trees are better protected in a phenological manner against such heavy infestations, as spring moth larvae begin to appear before their bud burst. The associational effects of EF trees on LF ones are only poorly known. We studied whether or not the infestation of LF trees by spring Lepidoptera can be affected by EF ones if they grow in the immediate vicinity. We compared spring assemblages of leaf-eating larvae of Lepidoptera on LF *Quercus cerris* L. with those on EF *Q. pubescens* Willd. in several microhabitats in Slovakia, Central Europe. Larvae were collected from mature and young trees. Mature trees sampled were growing: (1) in a closed-canopy forest; (2) in small groups; or (3) as a lone tree. Forest and tree groups are both constituted by oak species. Tree groups and lone trees were 20–50 m distant from forest edges. Young trees were growing (1) under mature *Q. pubescens* trees in a forest or (2) as a lone tree within forest gaps or near the edges. In the closed-canopy forest where LF trees (*Q. cerris*) were surrounded by EF ones (*Q. pubescens*), the caterpillars on mature LF trees were in abundance, almost as on mature EF ones. The species composition of larval assemblages on the two oak species was similar. In contrast, on small groups and on lone trees, the lepidopteran larvae were significantly less abundant on LF trees than EF ones. In the case of young trees, the abundance of larvae and their composition assemblages on both oaks were comparable in the forest. In the open habitat, LF trees were less infested by larvae than EF ones and the assemblages of moth larvae differed between the two. Our results reveal the effect (associational susceptibility) of EF trees on LF ones when growing in a close vicinity. It means that the phenological protection of LF trees may not be sufficient if they grow close to or are surrounded by EF ones.

**Keywords:** associational susceptibility; *Quercus*; forest protection; phenological synchrony; *Operophtera brumata*; *Agriopis leucophaearia*; bud burst; herbivory

## 1. Introduction

In temperate European forests, many early-flushing (EF) deciduous trees host numerous species of spring-feeding caterpillars of Lepidoptera, among them well-known *Operophtera brumata* (Linnaeus, 1758), *Erannis defoliaria* (Clerck, 1759), *Agriopis* spp., *Tortrix viridana* (Linnaeus, 1758), *Lymantria dispar* (Linnaeus, 1758), etc. [1–12]. They often cause defoliation of woody plants in forests or fruit trees.

The successful development of these Lepidoptera depends on synchrony between the hatching of caterpillars and the bud burst of host trees [13–21]. Neonates from folivorous caterpillars of early spring Lepidoptera have access to suitable food if they hatch or activate after overwintering inside the narrow phenological window occurring right after the bud burst of their hosts. If they hatch too early—before the bud burst, they starve because flushing buds and young foliage are lacking [14,19,22]. Spring caterpillars can resist starvation only for a few days [14]. On the other hand, if they do it too

late—relatively long after the bud burst, they also suffer from a lack of suitable food (i.e., young foliage) on host trees because the increased content of non- or low-digestible compounds and protective chemicals in maturing and mature leaves have a negative impact on caterpillar performance [19,23–29].

Caterpillars hatching early and late, which miss the window of opportunity for feeding on young foliage on the primary host tree (the one where they are born) need to locate food, usually on another tree. Baby caterpillars can achieve long-distance dispersal through ballooning—they use a silk thread or long setae on their bodies to float on air currents or to be borne by wind [30–38]. Caterpillars can leave unsuitable places also by lowering on a silk thread or dropping [31,39–41], or by walking from one tree to another where they touch each other [42].

The effects of woody plant diversity on herbivory in forests have often been studied with variable results [43–49]. Neighbouring trees can increase or decrease insect attacks to those that are at the centre (associational susceptibility or resistance) [50]. It is generally accepted that insect herbivory is lower in mixed forest stands than in pure stands, i.e., associational resistance e.g., [51–53]. However, many authors emphasise that beyond tree diversity *per se*, these effects are strongly dependent on the traits of host trees as well as the specificity of insect phytophages e.g., [51,53–57].

There is a little knowledge on fine-scale mechanisms leading to the associational effects of neighbouring plants on those that are at the centre [45,46,48,57–59]. For example, herbivores can move from one host plant to another neighbouring one [50]. There are few data on how EF trees affect the occurrence and abundance of spring caterpillars on late-flushing (LF) ones. Nealis and Régnière [60] recorded redistribution of late-instar caterpillars belonging to *Choristoneura fumiferana* Clemens, 1865, from damaged EF to undamaged LF host trees. Schafellner et al. [61] supposed that older caterpillars of *Lymantria dispar* left primary food places on *Quercus petraea* (Matt.) Liebl. trees and moved to young foliage on LF *Q. cerris* L. trees. Wesołowski and Rowiński [7] stated that LF oaks (*Quercus robur* L.) co-occurring with EF trees were visibly defoliated only during an outbreak.

Mixed deciduous forests comprising trees with various stages of bursting are widespread in temperate Europe [62–67]. We assume, therefore, that neighbourhood of early- and late-flushing co-occurring trees can significantly affect the infestation of the latter by spring Lepidoptera.

Studies on defoliators in fragmented forests or, in general insect herbivores in fragmented habitats, have brought conflicting results [47,68–71] which reflect the specific conditions of habitats (fragment size and quality, degree of insulation) and the characteristics of studied organisms (host plants and insect phytophages) [70,72–78]. To our knowledge, there is no study published about the impact of spring caterpillars, from EF trees on LF ones in small forest fragments.

Oaks (*Quercus* spp.) are among the most infested deciduous trees by early spring Lepidoptera in Central Europe [4,5,79,80]. From all oak species in this region, *Q. cerris* is the last one in terms of bud bursts [81] and, compared with other oaks, this delay is approximately two weeks [61,82]. The *Quercus cerris* often grows with EF deciduous woody species in mixed forests and is also frequent in forest fragments [66,83,84].

*Quercus cerris* and *Q. pubescens* Willd. are well adapted to moderate drought stress in summer, and due to a climate change, their increasing importance is expected in temperate European forests [85–91] as well as in Central Europe [92].

We studied (1) whether the infestation of mature LF trees by caterpillars can be affected by neighbouring mature EF ones in the forest interior and at a distance from it—in small mixed tree groups 20–50 m away from the edges, and (2) whether the infestation of young LF trees by caterpillars can be by neighbouring mature EF ones. We focused on early spring leaf-chewing caterpillars belonging to the group of "brumata-viridana complex" [10] on two co-occurring oak species—the LF Turkey oak (*Quercus cerris*) and the EF pubescent oak (*Q. pubescens*). The caterpillar assemblages, usually dominated by the well-known pests *Operophtera brumata* and *Tortrix viridana* (hence the name "brumata-viridana complex"),

comprise numerous lepidopteran species occurring first in spring and being synchronised with the bud burst of EF host trees [10].

We suppose that conditions for the development of neonate (first instar) caterpillars on *Q. cerris* are unfavourable due to late bud burst. Consequently, in early spring, the caterpillar abundance on LF *Q. cerris* would be lower than on EF *Q. pubescens.* Later, when the young leaves of *Q. cerris* unfold, caterpillars can switch their hosts to *Q. cerris* trees and profit from the food with higher nutritional quality occurring on this oak species [61]. The probability of *Q. cerris* being infected by caterpillars from *Q. pubescens* would be the highest in the case of trees in close neighbourhood belonging to both oak species [63,93]. We assume, therefore, that the caterpillar abundance on mature LF *Q. cerris* in a closed-canopy forest surrounded by EF *Q. pubescens* will be higher than the one on LF *Q. cerris* located out of the forest—in small mixed tree groups and solitary trees. Similarly, caterpillar assemblages on young LF (low) *Q. cerris* growing under the crowns of mature EF *Q. pubescens* are expected to be enriched by caterpillars from those mature crowns, unlike the young trees in open space [94]. Thus, mature and young LF trees that are more isolated from EF ones should be less infested by caterpillars.

Our study is clarifying one little-known aspect of a fine-scale mechanism leading locally to the tree diversity effects on herbivory cf. [48,59,74].

## 2. Materials and Methods

### 2.1. Studied Area

The research was carried out in Krupinská planina plateau (Southern Slovakia; 48°10′0.19″ N, 18°59′46.08″ E) in the southern part of the Western Carpathians Mts, at altitudes between 265 m and 330 m a.s.l. The study area belongs to a warm region with mean annual air temperatures 8–9 °C and mean annual precipitations 600–700 mm [95]. It is covered with xeric and thermophilous vegetation consisting of an oak forest and a forest-steppe. The share of *Quercus pubescens* in the forest was accounting for about two thirds of trees, and that of *Q. cerris* about one third. There are other tree species much less frequent and less abundant (up to 2%), such as *Acer campestre* L., *A. tataricum* L., *Tilia* spp., *Sorbus torminalis* (L.) Crantz., *Crataegus* spp., *Carpinus betulus* L., *Pyrus* spp. and *Ulmus* spp. The average height of mature oaks is 8–12 m and their age is about 70 years. Young trees and saplings are scattered in the forest understory. The adjacent forest-steppe constitutes a grassland with small tree patches and groups, and solitary trees. Like in the forest, EF *Q. pubescens* and LF *Q. cerris* trees are the most frequent here. Several shrub species such as *Rosa* spp., *Crataegus* spp., *Ligustrum vulgare* L. and *Prunus spinosa* L. also grow in tree patches. The study area was approximately 100 ha (50 ha of forest, 50 ha of forest steppe).

### 2.2. Data Collecting

Caterpillars were obtained from oaks by beating tree branches over a beating tray with 1.0 m diameter [96], i.e., samples originated from one-metre-long branches or one-metre-long terminal parts of them. Collections were carried out during the non-outbreak period in the first week of May in years 2015 and 2016.

In the first year, caterpillars were collected from mature LF *Q. cerris* and EF *Q. pubescens* growing in three types of microhabitats: (1) in a forest; (2) in small mixed tree groups (3–6 trees) composed of both oak species growing in a forest-steppe out of a dense forest; and (3) as solitary (lone) trees also out of it. Regarding the forest, caterpillars were sampled in the forest interior (canopy 80%) from mature trees located at a distance of 30–50 m from the edges, and at least 20 m from each other.

Every LF *Q. cerris* tree on which the caterpillars were collected was surrounded by EF *Q. pubescens* with crowns touching those of *Q. cerris* trees. Also crowns of *Q. cerris* within the mixed groups touched those of *Q. pubescens* trees. Mixed tree groups and solitary trees selected for this study were at a distance of 20–50 m from forest edges, other trees or groups. From each oak species in every microhabitat (i.e., forest, mixed tree groups, and lone trees), 11 samples of caterpillars were obtained, which are 66 (2 × 3 × 11) if taken together. One

sample represented caterpillars beaten from three branches (up to 3 m from the ground level), from one mature tree.

In the following year, caterpillars were collected from young LF *Q. cerris* and EF *Q. pubescens* (each tree up to 2 m high) growing in two types of microhabitats: (1) in the forest interior under the crowns of mature *Q. pubescens* at a distance of 30–50 m from the edges and (2) in open forest glades or edges, as solitary (lone) trees. These solitary young trees were located at 3–5 m from the crowns of mature trees belonging to any species, including *Q. pubescens*, and at a distance of at least 10 m from other young trees (Figure 1). From each oak species in every microhabitat (i.e., in a forest under the crowns of *Q. pubescens*, and out of forest in open glades or edges), we collected 15 samples of caterpillar assemblages, which are 60 ($2 \times 2 \times 15$) if taken together. One sample represented caterpillars beaten from two branches, from one young tree. Since the mixed groups of young trees composed of LF *Q. cerris* and EF *Q. pubescens* altogether were scarce in the study area, we did not include that kind of microhabitat in our research.

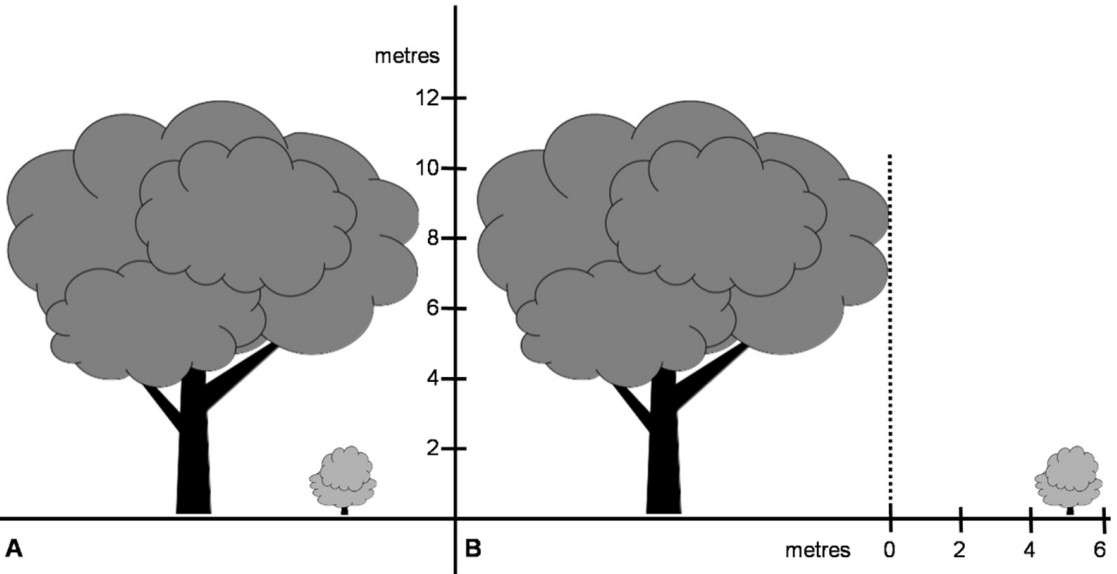

**Figure 1.** Position of young trees selected to sample caterpillars (mature trees—*Quercus pubescens*; young trees—*Q. cerris* or *Q. pubescens*; (**A**)—in the forest interior; (**B**)—in open forest glades or edges [as a solitary tree]).

Caterpillars were preserved in 75% ethanol and identified in the laboratory by using guides [2,82,97,98]. The nomenclature of lepidopteran species follows Pastorális et al. [99]. Two species, *Agriopis marginaria* (Fabricius, 1776) and *A. aurantiaria* (Hübner, 1799), which caterpillars are difficult to distinguish from each other according to their external morphology, were considered as a single taxon in the analyses.

### 2.3. Data Analyses

The caterpillar total abundance, and the abundance of *A. leucophaearia* (Denis and Schiffermüller, 1775) and *O. brumata*, were statistically compared using analysis of deviance (ANODEV), for mature and young LF *Q. cerris* and EF *Q. pubescens* separately. The ANODEV model with a negative binomial error distribution and a log-link function was used to test the effect of tree species and forest structural combinations on the abundance. Permutational multivariate analysis of variance (perMANOVA) was employed for testing the effect of tree species and forest structural combinations on the composition of species assemblages [100]. Data on species abundance were log(x + 1) transformed and the Bray-Curtis dissimilarity index [101] was used. The results were presented using the non-metric multidimensional scaling ordination technique (NMDS) [102].

The significant level of 0.05 was applied. Statistical analyses and graphical outputs were made in R [103] package boot [104], also using ggplot2 [105], MASS [106], multcomp [107] and vegan [108].

## 3. Results

### *3.1. Caterpillars on Mature Trees*

*Abundance* (Figure 2). Caterpillars on LF *Q. cerris* were abundant only in the forest interior and this differed significantly from that on *Q. cerris* in mixed tree groups (z = −4.280, *p* < 0.001) and on lone trees (z = −4.796, *p* < 0.001). In contrast, caterpillars on EF *Q. pubescens* were in abundance in all microhabitats without differences between forest and non-forest environment. A difference between *Q. cerris* and *Q. pubescens* within the forest was considerable but not significant (z = 2.413, *p* = 0.088).

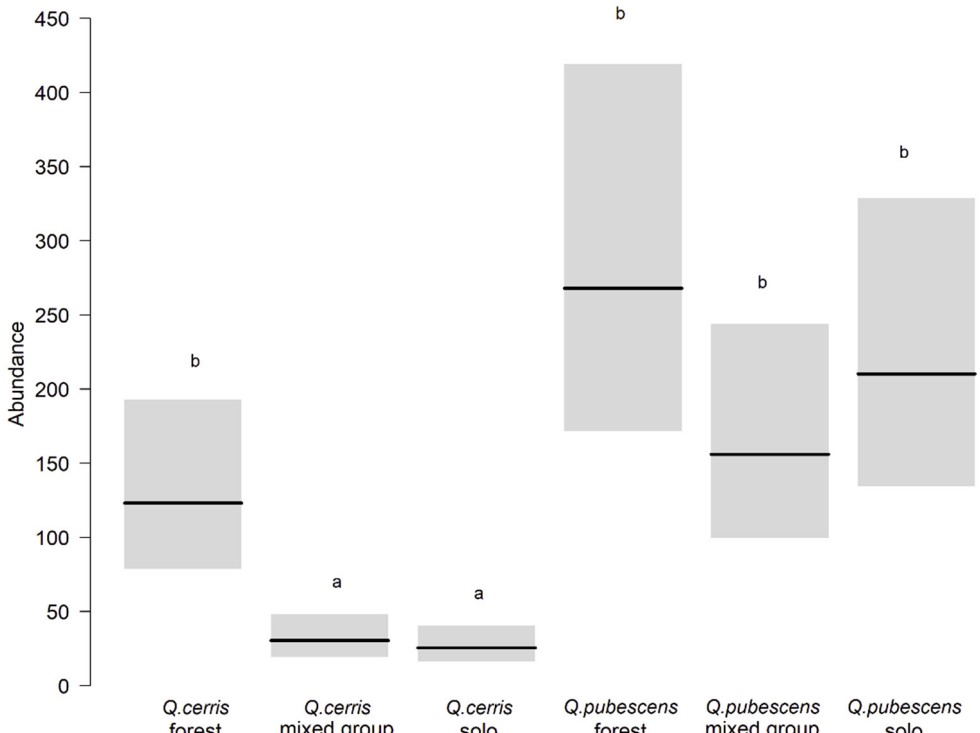

**Figure 2.** Abundance of caterpillars (number of individuals on three branches, each 1 m long) on mature LF *Quercus cerris* and EF *Q. pubescens* located in the forest interior (forest) and out of it, in mixed tree groups composed of *Q. cerris* and *Q. pubescens*, and on lone (solo) trees. A horizontal line denotes the mean, and bars the 95% confidence intervals. Distinct letters above columns indicate a significant difference.

*Species composition* (Figure 3). Caterpillar assemblages on LF and EF trees did not differ between each other in the forest interior (F = 2.094, *p* = 0.105) but they significantly did in a non-forest environment—in mixed tree groups (F = 11.258, *p* < 0.001) and on lone trees (F = 12.639, *p* < 0.001). Assemblages on *Q. cerris* in microhabitats out of the forest had a similar composition (F = 0.487, *p* > 0.100) but they were different from those in forest (mixed groups: F = 4.660, *p* = 0.003; lone trees: F = 5.836, *p* < 0.001). The same was found for assemblages on *Q. pubescens* (mixed groups vs. lone trees: F = 2.104, *p* > 0.05; mixed groups vs. forest: F = 4.331, *p* = 0.001; lone trees vs. forest: F = 4.709, *p* < 0.001).

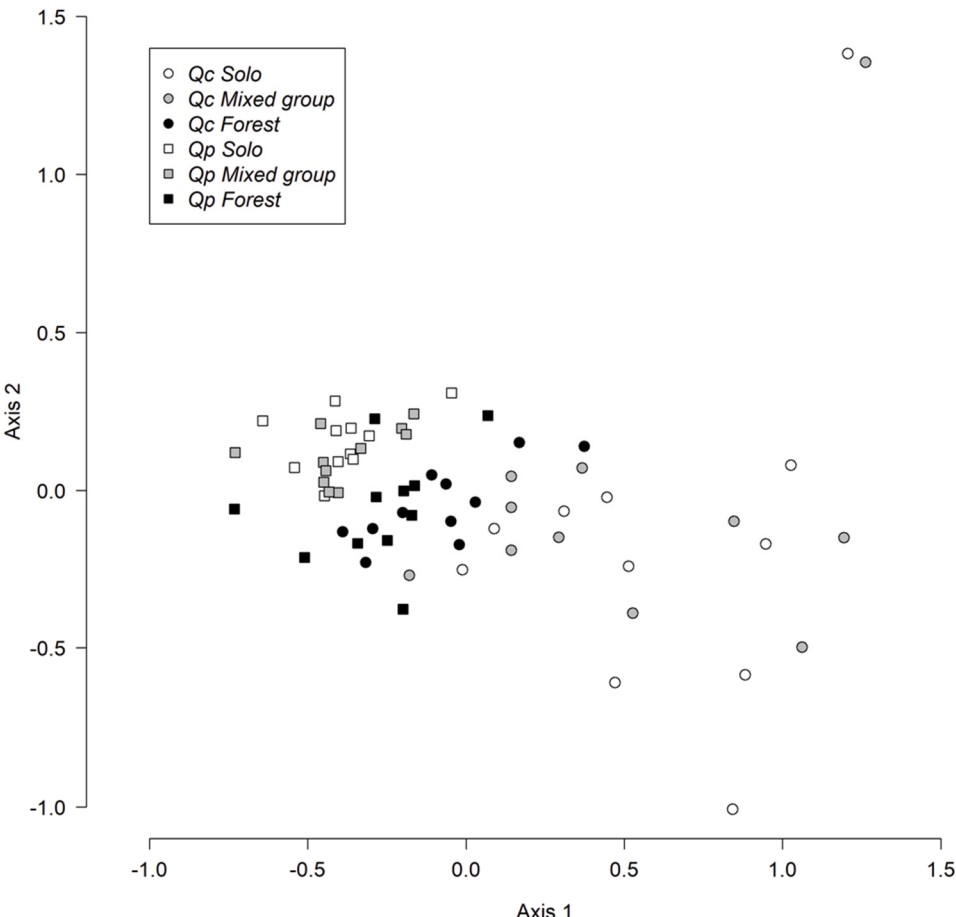

**Figure 3.** Non-metric multidimensional scaling of caterpillar assemblages on mature LF *Quercus cerris* and EF *Q. pubescens*. (Qc—*Q. cerris*, Qp—*Q. pubescens*, Solo—lone trees, Mixed group—mixed tree groups composed of *Q. cerris* and *Q. pubescens*, Forest—trees growing in the forest interior).

*Agriopis leucophaearia* (Figure 4). This species was predominant in all microhabitats on both oak species (LF *Q. cerris*—dominance in forest: 87%, in mixed tree groups: 72%, on lone trees: 78%; EF *Q. pubescens*—in forest: 91%, in mixed groups: 72%, on lone trees: 69%), significantly influencing the abundance of caterpillar assemblages. On *Q. pubescens*, this moth was abundant in all microhabitats. In contrast, on *Q. cerris* it was abundant only in the forest interior and much less abundant in mixed tree groups ($z = -4.111$, $p < 0.001$) and on lone trees ($z = -4.342$, $p < 0.001$).

*Operophtera brumata* (Figure 5). The second most abundant moth also appeared on the two oak species (LF *Q. cerris*—dominance in forest: 4%, in mixed tree groups: 2%, on lone trees: 2%; EF *Q. pubescens*—in forest: 3%, in mixed groups: 11%, on lone trees: 19%). Its caterpillars on both oaks were similarly abundant in the forest interior ($z = 1.577$, $p > 0.100$). On *Q. cerris,* they were significantly less in abundance out of it—in mixed tree groups ($z = -3.708$, $p < 0.001$) and on lone trees ($z = -4.072$, $p < 0.001$). In contrast, on *Q. pubescens*, they were more abundant in mixed tree groups ($z = 2.621$, $p = 0.034$) and especially on lone trees ($z = 5.258$, $p < 0.001$) when compared with the forest interior.

### 3.2. Caterpillars on Young Trees

*Abundance* (Figure 6). Caterpillars were abundant on both oak species growing under mature EF *Q. pubescens* in the forest interior and did not differ significantly between each other ($z = 1.275$, $p > 0.100$). Those on LF *Q. cerris* were less abundant on lone trees than in the forest interior ($z = -3.520$, $p = 0.002$) while caterpillars on EF *Q. pubescens* were also in abundance on lone trees, and this did not differ from that within the forest ($z = -1.296$, $p > 0.100$).

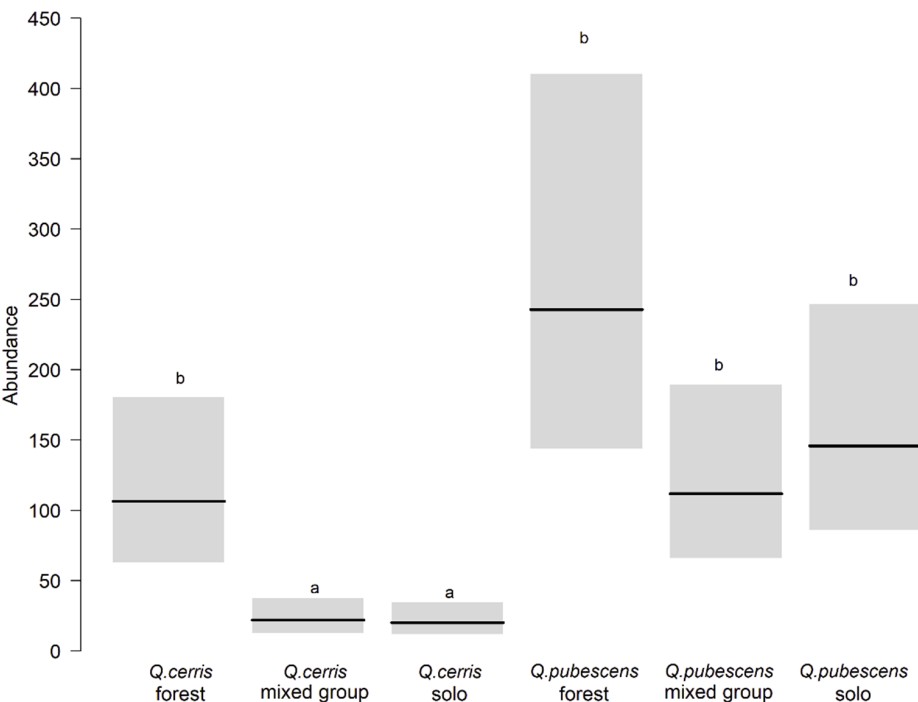

**Figure 4.** Abundance of *Agriopis leucophaearia* caterpillars (number of individuals on three branches, each 1 m long) on mature LF *Quercus cerris* and EF *Q. pubescens* located in the forest interior (forest) and out of it, in mixed tree groups composed of *Q. cerris* and *Q. pubescens*, and on lone (solo) trees. A horizontal line denotes the mean, and bars the 95% confidence intervals. Distinct letters above columns indicate a significant difference.

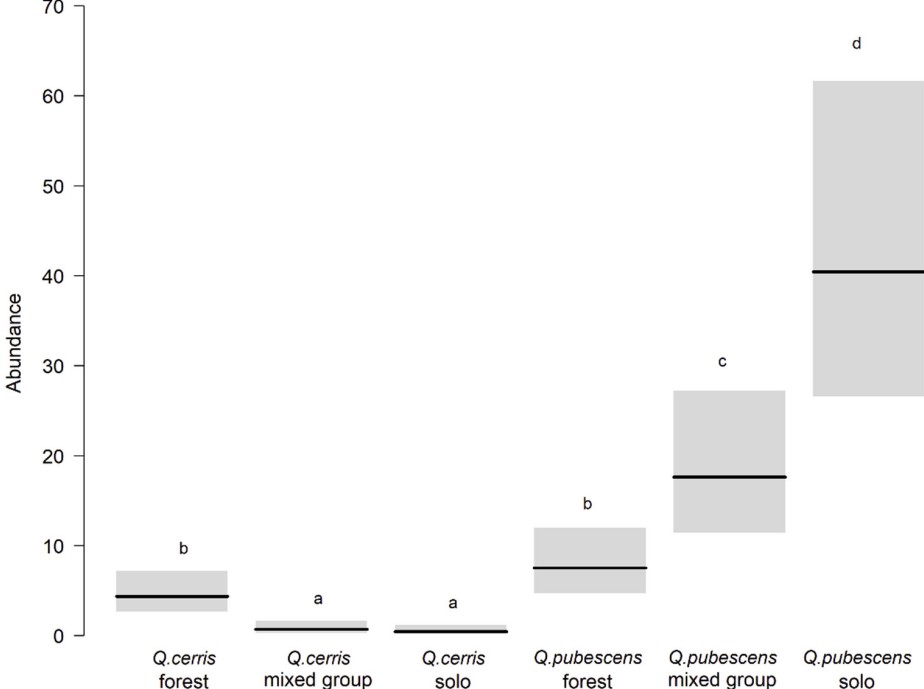

**Figure 5.** Abundance of *Operophtera brumata* caterpillars (number of individuals on three branches, each 1 m long) on mature LF *Quercus cerris* and EF *Q. pubescens* located in the forest interior (forest), and out of it, in mixed tree groups composed of *Q. cerris* and *Q. pubescens*, and on lone (solo) trees. A horizontal line denotes the mean, and bars the 95% confidence intervals. Distinct letters above columns indicate a significant difference.

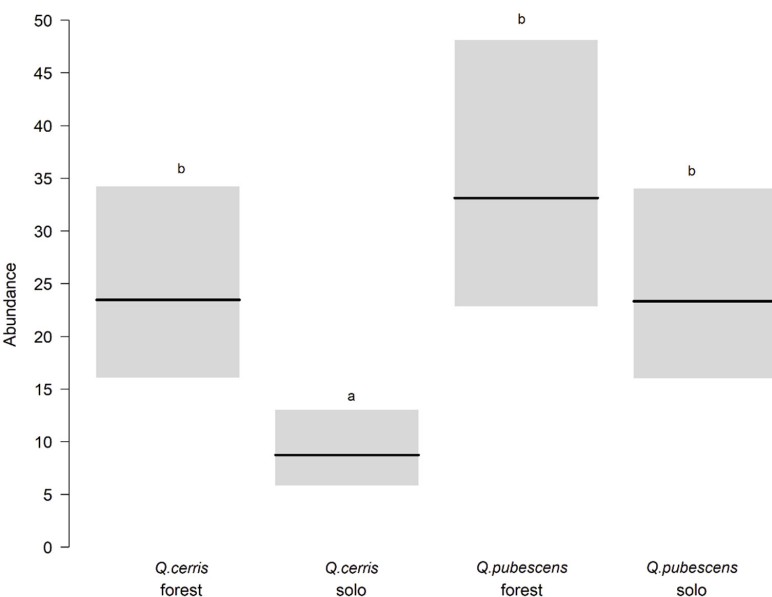

**Figure 6.** Abundance of caterpillars (number of individuals on two branches, each 1 m long) on young LF *Quercus cerris* and EF *Q. pubescens* located in the forest interior and on lone (solo) trees in open forest glades or edges. A horizontal line denotes the mean, and bars the 95% confidence intervals. Distinct letters above columns indicate a significant difference.

*Species composition* (Figure 7). Caterpillar assemblages on LF *Q. cerris* and EF *Q. pubescens* did not differ between each other in the forest interior (F = 2.586, $p$ = 0.082) but they significantly did in a non-forest environment, i.e., on lone trees (F = 3.100, $p$ = 0.014). In the case of *Q. cerris*, assemblages on lone trees were different from those within the forest (F = 3.136, $p$ = 0.014) while for *Q. pubescens*, they were similar on lone trees and in the forest interior (F = 0.778, $p$ > 0.1).

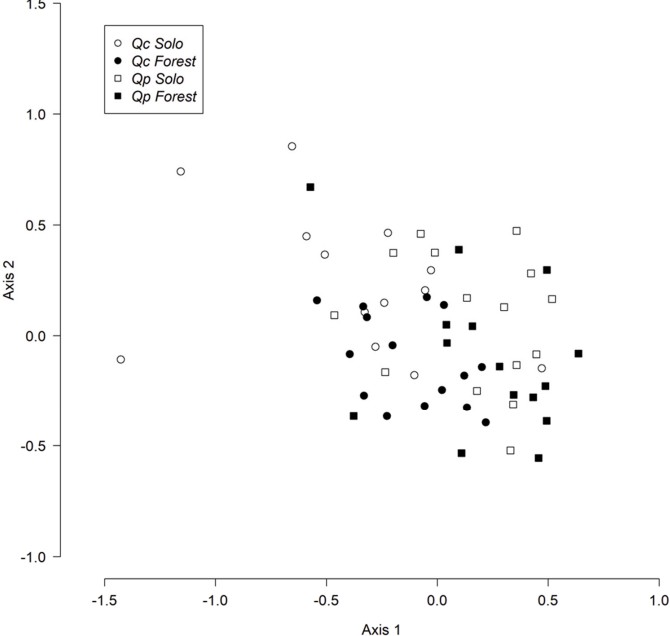

**Figure 7.** Non-metric multidimensional scaling of caterpillar assemblages on young LF *Quercus cerris* and EF *Q. pubescens*. (Qc—*Q. cerris*, Qp—*Q. pubescens*, Solo—lone trees growing in open forest glades or edges out of the crowns of mature *Q. pubescens*, Forest–trees in the forest interior under the crowns of mature *Q. pubescens*).

*Agriopis leucophaearia* (Figure 8). It was a predominant species also on young LF *Q. cerris* (dominance in the forest interior: 83%, lone trees: 77%) and EF *Q. pubescens* (within the forest: 70%, lone trees: 67%). In the forest interior, there was a non-significant difference between caterpillar abundances on *Q. cerris* and *Q. pubescens* (z = 0.559, *p* > 0.100). On lone trees, their abundance on *Q. cerris* was significantly lower than that in the forest interior (z = −3.390, *p* = 0.005) while on *Q. pubescens* it was relatively high, and the difference between abundances, in both microhabitats, was not significant (z = −1.309, *p* > 0.100).

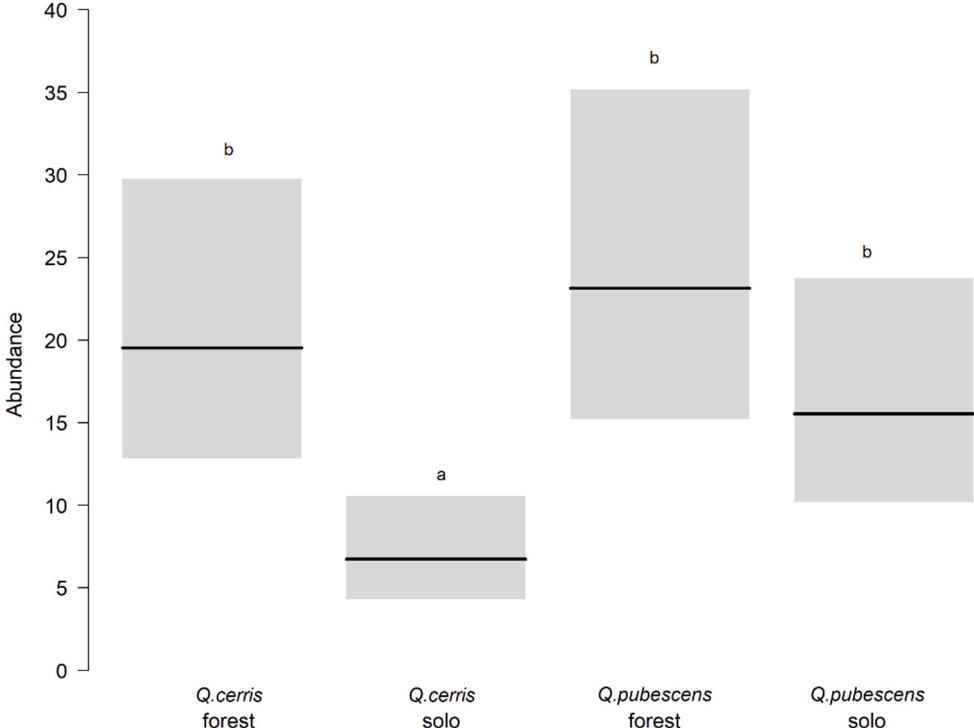

**Figure 8.** Abundance of *Agriopis leucophaearia* caterpillars (number of individuals on two branches, each 1 m long) on young LF *Quercus cerris* and EF *Q. pubescens* located in the forest interior and on lone (solo) trees in open forest glades or edges. A horizontal line denotes the mean, and bars the 95% confidence intervals. Distinct letters above columns indicate a significant difference.

## 4. Discussion

Our research has revealed that caterpillar assemblages on LF *Q. cerris* and EF *Q. pubescens* were similar in the closed-canopy mixed forest composed of both LF and EF trees and different on those growing out of it. In the forest, the caterpillar abundance on mature LF trees almost reached that on mature EF trees (insignificant difference). In microhabitats out of the forest—on small mixed tree groups composed of both species and on lone trees, caterpillars on LF oaks were significantly less abundant than on EF ones. Moreover, the species composition of their assemblages on LF *Q. cerris* and EF *Q. pubescens* was similar in the forest interior but different in microhabitats out of it. It suggests an increased infestation by caterpillars (measured as a caterpillar abundance) of LF trees surrounded by EF ones in forest (associational susceptibility). This latter effect could be the result of a spill-over where herbivores move from neighbouring primary host trees (*Q. pubescens*) onto secondary host trees (*Q. cerris*) at the centre.

Early spring Lepidoptera, as recorded, can develop on the leaves of both studied oak species [5,9,10,79,109,110], but caterpillars in Central Europe hatch synchronously with opening buds of EF *Q. pubescens* [5]. Previous studies have reported low abundances of caterpillars on *Q. cerris* [9,10,31,111] and J. Liška, pers. comm. These low abundances observed without taking into account the possible effect of neighbouring trees may reflect environmental conditions which are harsher for first-instar caterpillars on this LF

oak species than on other oaks. Our results have also confirmed the significantly lower abundance of caterpillars on LF *Q. cerris* than on EF *Q. pubescens* for solitary growing trees.

We studied the associational effect of neighbouring trees on spring caterpillar assemblages feeding on two oak species. The abundance and composition of these assemblages were mostly determined by two dominant moth species, *A. leucophaearia* and *O. brumata*, occurring frequently and in abundance on Central European oak species [2–5,9–12]. As for other tree species and their caterpillar assemblages, further research is needed.

Females of many early spring Lepidoptera oviposit at times different from those over which their offspring develops in. It means that these females do not experience environmental conditions their caterpillars are going to live next spring. Thus, there is a poor chance for them to select the best food resources for their future offspring [112]. Non-selective oviposition has been documented in common geometrids *O. brumata* and *Epirrita autumnata* (Borkhausen, 1794) [36,113,114]. Also, Tiberi et al. [115] recorded a similar number regarding eggs of *Tortrix viridana* on two oak species, *Q. cerris* and *Q. pubescens*. The oviposition of remaining species from the "brumata-viridana complex" has not been studied yet, but it is highly probable that the females place their eggs on both oaks.

The majority of caterpillars that hatched on LF oaks need to find new feeding places. Only some neonates among them can stay on this host tree—either those hatching later or those which are lucky to do it on a branch or tree with unusually early flushing buds. Previous studies suggest that small caterpillars leave relatively often their primary places to feed [18,31,32,113,116,117], however, inclination to disperse is a species-specific trait [14]. Baby caterpillars that were dispersing through ballooning can reach neighbouring or close-growing trees quite easily [31,113], and can increase caterpillar abundance on EF trees. The question of an extent to which neonate caterpillars being dispersed from LF *Q. cerris* affect neighbouring EF *Q. pubescens* trees remains still unknown. Similarly, the knowledge of dispersing late-instar caterpillars between trees is insufficient; this phenomenon has been most studied marginally, and only in some species so far [60,61,93,118,119].

There are plenty of abiotic and biotic stimuli inducing caterpillars to leave their feeding places (e.g., low quality or lack of food, a contact with a predator or parasitoid, physical stimuli caused by weather conditions, etc.) [4,60,61,94,120–126]. Many caterpillars can be seen as they climb tree trunks to get into crowns after rainstorms or strong winds (pers. observations of authors). Dispersing larvae have a better chance to survive when finding suitable feeding places near their primary host trees. It is assumed that the impact of those caterpillars on other trees is predominantly local [63,93,127,128].

The presence of EF *Q. pubescens* in isolated small mixed tree groups seems to be insufficient to increase considerably the caterpillar abundance or affect the composition of their assemblages on LF *Q. cerris*. Dispersal of neonate caterpillars (at least some species) may easily overcome distances between microhabitats we have studied (20–50 m) ([30–33] and references therein). However, ballooning is of little importance for the redistribution of caterpillars onto *Q. cerris* trees as their buds are still mostly closed at a time when they are hatching. Older caterpillars have limited ability to spill over, from EF primary host to neighbouring LF trees growing in small isolated groups. Hanging on silk threads or falling on the ground in such a microhabitat, they may easily miss neighbouring trees, and get lost in open space. Similarly, we suppose that solitary growing trees are highly improbable to be reached by older caterpillars from a forest or other trees being several tens of metres away. Consequently, in microhabitats out of forest, adults that originated from caterpillars living on LF *Q. cerris* are less abundant than those on EF *Q. pubescens*. Thus, the abundance of females laying eggs on *Q. cerris* is also lower and only a small part of offspring (i.e., caterpillars having hatched later) complete its development on trees belonging to this LF oak species. To summarise it, the small number of eggs and the few possibilities to enrich their assemblages from more infested EF *Q. pubescens* contribute together in low caterpillar abundances on LF *Q. cerris* trees.

Very low caterpillar abundance on LF *Q. cerris* in small fragments distant several tens of metres from a continuous forest suggests that no specialised lepidopteran population is

genetically adapted on these trees unlike some known cases [18,129,130]. In mixed forests where moth adults and caterpillars move between trees, interbreeding of individuals that develop on different plants occurs. Both sexes or at least males, and some neonate caterpillars can overcome short distances between a forest and its fragments [31,131,132] is what inhibits a genetically determined specialisation on trees with specific phenology [19].

The caterpillar abundance on mature EF *Q. pubescens* in small forest fragments, i.e., mixed tree groups and solitary trees, almost reached that in the forest interior but the composition of their assemblages in these microhabitats differed from those in forest. On oaks, a small or positive effect of forest fragmentation on herbivory has been recorded e.g., [47,77]. Caterpillars living on solitary trees and those in small groups out of forest are influenced by distinct conditions present within the forest [133–136] as well as the different quality of leaves, as their food plays an important role when compared with these in the forest interior [137–140]. The above-mentioned conditions together with the preference of certain habitats varying among species [5,141–143] determine the assemblages of herbivores.

*Agriopis leucophaearia*, the most dominant species recorded in caterpillar assemblages, on mature EF *Q. pubescens*, was in abundance, almost as in microhabitats out of forest as within it (differences were not significant), thus, it follows that the moth does not prefer any of the specific environments given. This species was abundant as well on *Q. pubescens* in open-canopy forest [10]. In contrast, another dominant species, *O. brumata*, appeared on *Q. pubescens* in higher abundance in microhabitats out of forest than in a continuous forest. It could be caused by its different habitat preference, since this moth is also abundant in fruit orchards [93,113,114,144–147], parks and urban alleys of trees [142,148,149] where the latter or shrubs do not grow close to each other. Van Dongen et al. [68] and van Dongen & Scott [72] studied *O. brumata* in patches, larger and more isolated in comparison with those in our research, and they recorded the negative effect of patch isolation on this moth.

On mature LF *Q. cerris*, caterpillar assemblages on small tree groups and on solitary trees differed from those in the forest interior. Although *A. leucophaearia* and *O. brumata* were dominant on these oaks in the studied small forest fragments, being very low abundant suggest that *Q. cerris* is not a suitable host for them in such microhabitats. Other lepidopteran species occurred there in low abundance, too.

We have also recorded associational susceptibility in young LF *Q. cerris* growing in close vicinity (i.e., right under the crowns) of mature EF *Q. pubescens* within the forest. These young LF trees were infested by caterpillars more significantly than young ones in open space—in open forest glades or edges. In forest, caterpillar assemblages on young LF and EF trees were similar. These results suggest that a close distance between young LF trees and mature EF ones is crucial because dispersing caterpillars reach rather closely trees in growth. Saplings and other plants in the forest understory are known to be infested by caterpillars that descend on silk threads or fall from the forest canopy (e.g., in searching for food) [5,39,94,121,141]. The frequency of the movements made by caterpillars between forest strata has not been studied so far. The reported abundance of the dominant species *A. leucophaearia* suggests that young LF *Q. cerris* growing out of the close range from mature EF *Q. pubescens* be less suitable than trees in forest under the crowns of mature *Q. pubescens*, for the larval development of this moth. We did not statistically analyse other lepidopteran species due to their low abundances.

Effects of associational susceptibility on secondary host trees are known mainly during outbreaks when previously unsuitable or suboptimal hosts were also infested [4,7,45,60,150]. We have recorded positive associational effects of EF trees on LF ones at reduced (non-outbreak) herbivore abundance. Our results show that the effect of EF trees on LF ones is manifested only in close vicinity of trees in forest. So, tree density and forest fragmentation can modify the strength of the associational effect trees with different phenology have. The high caterpillar abundance on LF *Q. cerris* in closed-canopy forest seems to be the result of the tree-to-tree movement of older larvae and probably of high numbers of moth eggs laid on *Q. cerris* (comparable with those on *Q. pubescens*). Also, other studies describing the

effects of associational susceptibility on woody plants are linked to forests or dense stands of trees [4,7,39,45,60,61,94,150,151]. The importance of the close vicinity around early- and late-flushing trees for associational effects was indicated by the results of our previous research. In a sparse forest (an open-canopy forest) dominated by both studied oak species, mature and young LF *Q. cerris* were significantly less infested by early spring caterpillars than EF *Q. pubescens* [10].

The composition of caterpillar assemblages on both oak species was similar in forest but differed in microhabitats out of it. It suggests that the associational effect of EF oaks on LF ones is related to many lepidopteran species. The most abundant moth, *A. leucophaearia*, can feed on broad-leaved tree species preferring oaks [10,80,152]. The second in abundance, *O. brumata*, is a typical generalist [31,80,82,114]. The studied type of associational susceptibility can develop when both plants at the centre and in the surroundings are palatable for these herbivores [50]. In case of phylogenetically related and palatable hosts (in our study, *Q. cerris* and *Q. pubescens*), herbivores can have a broad [4,51,150] as well as a narrow diet breadth [45,60].

Except for well-known species of early spring Lepidoptera that are considered pests, *A. leucophaearia* should also be added to the list of forest pests, since it was abundant in our study area and other Central European regions, too [10,12,82,153,154].

The method used (branch beating) is appropriate and advantageous for collecting externally feeding leaf-chewing caterpillars of early spring Lepidoptera [155]. In case of mature trees, these were only obtained from the lower part of tree crowns (up to 3 m from the ground) but for the comparison of caterpillar assemblages between two oak species it was sufficient. All comparable microhabitats in this study were located in the same biogeographical area and landscape structure. In each microhabitat examined, the same (or very similar) abiotic (e.g., climatic) and biotic (e.g., predators and parasitoids) factors acted on caterpillars on both oak species (*Q. cerris* and *Q. pubescens*). There were differences only between conditions associated with host species (different phenology and food quality). This enabled us to acquire new knowledge on fine-scale mechanisms leading to the increased caterpillar infestation of LF trees in the immediate vicinity of EF ones.

Our results suggest that EF trees influence the surrounded LF ones regularly (every season). Such influence can be modified by other factors. For example, the content of chemical substances in oak leaves is species-specific [156,157], so we cannot rule out differences in the palatability of *Q. cerris* and *Q. pubescens* leaves for early spring caterpillars. Severe defoliation can induce resistance against leaf-chewing insects in the following year [158]. On the other hand, trees which usually few caterpillars feed on, i.e., those with low resistance (such as *Q. cerris*) could be sporadically heavily infested by herbivore insects, for example by *Lymantria dispar* L. [159]. In addition, global warming can disrupt the synchrony of EF oak and spring Lepidoptera phenology [160] and affect the abundance of caterpillars on trees. Associational effects of EF trees on LF ones can be connected with forest health deterioration, as weakened trees are often infested by various insect pests [5]. Moreover, young leaves produced by damaged oaks are often damaged by oak powdery mildew [161]. Further studies should address these issues in detail.

Our findings are important for silvicultural practices. The establishment of mixed forest stands with distinct tree species or cultivars using a different time for bursting can increase the susceptibility of certain trees to the attack of herbivores. These trees can be considered resistant if growing in monocultures or with other species with similar phenology. For example, in monocultures of LF *Q. cerris* in Central Europe, Lepidoptera within the "brumata-viridana complex" occur in small abundances (J. Liška, pers. comm.). The seedlings of resistant species or cultivars planted under mature trees with different phenology may also be attacked by herbivores more strongly than when they are in open space. Before the establishment of any forest or park stand, we recommend, therefore, to take into account traits of trees and the local fauna of potential pests.

**Author Contributions:** Conceptualization, L.S., P.Z. and J.K.; methodology, L.S., P.Z. and J.K.; software, M.P.; validation, L.S., M.P. and J.K.; formal analysis, L.S., M.P. and M.S.; investigation, L.S., P.Z., M.P., M.S. and J.K.; resources, L.S. and J.K.; data curation, L.S.; writing—original draft preparation, L.S. and J.K.; writing—review and editing, L.S., P.Z., M.P., M.S. and J.K.; visualization, L.S. and M.P.; supervision, J.K.; project administration, J.K. and P.Z.; and funding acquisition, J.K. and P.Z. All authors have read and agreed to the published version of the manuscript.

**Funding:** This research was funded by the Slovak Grant Agency for Science (VEGA) via grant no. VEGA 2/0032/19 as well as by the Slovak Research and Development Agency (APVV) via grant no. APVV-19-0119.

**Acknowledgments:** We thank Dominique Fournier for linguistic improvements and Milan Mikuš for technical assistance with data collecting.

**Conflicts of Interest:** The authors declare no conflict of interest.

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
