# Peer review of "Infestation of Early- and Late-Flushing Trees by Spring Caterpillars: An Associational Effect of Neighbouring Trees"

_forests, doi:10.3390/f12091281_

Round 1

Reviewer 1 Report

The manuscript Infestation of early- and late-flushing trees by spring caterpillars: an associational effect of neighbouring trees reveals an important question of present interest. Knowledge of the Lepidoptera defoliators and their influence on deciduous forests is an interesting scientific subject and would be useful for experts working on the same pest problems in other countries in Europe.

All paragraphs in the manuscript correspond to the Forests Journal requirements. They are sufficiently informative and covering all sections - abstract, purpose, materials and methodological approach, results, and discussion.

Author Response

Dear reviewer,

Thank you very much for your evaluation of our manuscript.

The manuscript has been linguistically reviewed by a professional native speaker Dominique Fournier, M.Sc., Linguistic Services DF, Canada. www.serviceslinguistiquesdf.com,

info@serviceslinguistiquesdf.com

Best regards

Lenka Sarvašová, Ján Kulfan, Peter Zach

Reviewer 2 Report

Dear authors
would you like to comment on the following, for example, in the discussion: as you have written, there are great differences in the spring development of oaks (even several weeks between praecox and tardiflora forms). The earlier form is usually more affected by foliar pests [Oszako, T., & Delatour, C. (2000). Recent advances on oak health in Europe. Forest Research Institute], which has also been confirmed by your research. Therefore, from the point of view of the health of an oak stand, is it advisable to collect acorns under oaks of late varieties, as they are less damaged by insects? Or rather, as in the past, to collect acorns of the so-called native oak, which includes both varieties and, in addition, different oak species eg. Q. robur and Q.petraea. There is also a difference between them in the attractiveness of the leaves to insects. The latter species contains more bitter tannins and is therefore less attacked by primary pests. It also roots deeper and has a higher root biomass, making it more tolerant of prolonged droughts. These choices may be important to the breeding objective of maintaining forest sustainability. Are there similar relationships between Q. cerris and Q. pubescens?
Do the insects in the interaction respond to oak leaf development and adapt, for example, by delaying their development to wait for tree leaves (late-flushing LF)? Have oaks been observed to warn each other in situations of insect oubreak? Do heavily defoliated oaks send signals (e.g. volatiles or via fungal roots forming symbiotic associations - mycorrhizae)? Do they activate the synthesis of phenolic compounds as a defence mechanism or do they reduce foliar nutrition (less fatty acids) so that the feeding larvae are malnourished and susceptible to fungal and bacterial attack? 

If damaged oaks produce new leaves, are they infested with oak powdery mildew? New leaves are susceptible and their emergence from dormant buds coincides with sporulation of, for example, Erisyphe alphitoides. This fungus causes (especially in warm years) repeated defoliation, especially in Q. robur. Does this phenomenon also apply to Q. cerris and Q. pubescens?

Is the damage from folivorous caterpillars greater when they first appear? Do the trees respond if they return every year, and is the damage less as a result? Therefore, should foliage protection (chemical, biological) be applied to stands that are infested for the first time?

Does climate change (global warming) affect earlier leaf and insect emergence?Do falling insect extremes stimulate leaf development in subsequent years? Perhaps leaves are more nutrient-rich as trees have received organic fertilizer? This may also stimulate the development of foliar fungi?

Author Response

Dear reviewer,

Thank you very much for your comments and recommendation on our manuscript.

Thank you for your valuable suggestions and ideas for future reasearch, too. Your relevant comments have been incorporated into the manuscript (in the Introduction and mainly in the Discussion).

The manuscript has been linguistically reviewed by a professional native speaker Dominique Fournier, M.Sc., Linguistic Services DF, Canada. www.serviceslinguistiquesdf.com,

info@serviceslinguistiquesdf.com

Best regards

Lenka Sarvašová, Ján Kulfan, Peter Zach

Reviewer 3 Report

Interesting paper. A couple of notes:

  • All the scientific names (insects, plants...), when reported for the first time in the text should be written in full with Authority and systematics
  • an experimental design could be useful for a better understanding of the experiments

Author Response

Dear reviewer,

Thank you very much for your comments and recommendation on our manuscript.

We have completed all scientific names with Authority.

We have improved the Methods and made it more clear as you recommended.

The manuscript has been linguistically reviewed by a professional native speaker Dominique Fournier, M.Sc., Linguistic Services DF, Canada. www.serviceslinguistiquesdf.com,

info@serviceslinguistiquesdf.com

Best regards

Lenka Sarvašová, Ján Kulfan, Peter Zach

Reviewer 4 Report

The manuscript describes experiments designed to look at the influence of local habitat (number/density of neighbouring trees) on the number of lepidopteran herbivores on early and late flushing trees. The study is well designed and the results are clearly presented and contribute to our understanding of occurrence of at least some important herbivores. There are two major concerns that the authors should address to a larger extent.

The first is the question if striking effects of neighbouring EF Q. pubescens is a species effect (Q. pubescens hosting a large number of herbivore species and individuals) or an effect of “neighbouring”. Would the effect have been similar or different with other species of neighbouring trees? For instance, is the increased number of caterpillars on “Q. cerris forest” in Fig. 2 an effect of the habitat (inside) or of presence of Q. pubescens? Unfortunately, this has to be answered in a future study, but think it is important to elaborate on this in order to not generalize too much based on the presented investigation.

The second concern is about the large dominance of one or possibly two herbivorous species. It would be valuable to know how dependent the results are on these species. It would also be interesting to know if this dominance of one-few herbivorous species is the normal situation for these oak trees.

In addition, there are a number of minor issues to consider (by line number):

52: I usually understand “primary host” as the generally preferred one by a species. In this sentence, it is more a description of where the eggs/larvae happened to be found first.

95: Instead of “and out of it”, I would say “and at a distance from it”.

170: space after figure legend.

301: Please clarify what “These last abundances” means.

323: The sentence beginning with “The question” has to be reformulated into something understandable.

358-361: This is also a sentence that needs a revision to make sense.

545, 693: Species names in italics!

Figs. 2, 4 and 5 have the same categories (x-axis) and I suggest to merge the tree into one figure and decrease the height of each. Similarly, I suggest to merge Figs. 6 and 8.

Finally, are really all these 155 references needed?

Author Response

Dear reviewer,

Thank you very much for your comments and recommendation on our manuscript.

We accept your comments on our results based on caterpillar assemblages living on oaks and two dominant moth species. We have answered your questions in the Discussion.

We have corrected errors you mentioned in your review.

We have consulted sentences in the lines 323 and 358-361 with the native speaker and we will not change them.

The issue we researched is relatively wide covering several scientific areas (entomology, dendrology, phenology, behavioral ecology, forestry, etc.) which needs a rich base of literary resources cited in the manuscript. Moreover, we have supplemented several references as one of reviewers (the reviewer no. 2) requested.

The manuscript has been linguistically reviewed by a professional native speaker Dominique Fournier, M.Sc., Linguistic Services DF, Canada. www.serviceslinguistiquesdf.com,

info@serviceslinguistiquesdf.com

Best regards

Lenka Sarvašová, Ján Kulfan, Peter Zach